# Indoor Location Technology with High Accuracy Using Simple Visual Tags

**DOI:** 10.3390/s23031597

**Published:** 2023-02-01

**Authors:** Feng Gao, Jie Ma

**Affiliations:** College of Mechanical and Vehicle Engineering, Chongqing University, Chongqing 400044, China

**Keywords:** indoor location, visual location, error analysis, weighted least squares

## Abstract

To achieve low-cost and robustness, an indoor location system using simple visual tags is designed by comprehensively considering accuracy and computation complexity. Only the color and shape features are used for tag detection, by which both algorithm complexity and data storage requirement are reduced. To manage the nonunique problem caused by the simple tag features, a fast query and matching method is further presented by using the view field of the camera and the tag azimuth. Then, based on the relationship analysis between the spatial distribution of tags and location error, a pose and position estimation method using the weighted least square algorithm is designed and works together with the interactive algorithm by the designed switching strategy. By using the techniques presented, a favorable balance is achieved between the algorithm complexity and the location accuracy. The simulation and experiment results show that the proposed method can manage the singular problem of the overdetermined equations effectively and attenuate the negative effect of unfavorable label groups. Compared with the ultrawide band technology, the location error is reduced by more than 62%.

## 1. Introduction

With the development of artificial intelligent technologies, intelligent mobile platforms are becoming more widely used in the fields of healthcare, warehousing, logistics, industrial production, etc. [1,2]. Indoor location with high accuracy, as one of the key technologies, is the basis for autonomous functions such as navigation and decision. Compared with other application scenarios, it has special requirements on computation complexity, convenience of deployment, and robustness to environmental disturbances. Outdoors, as a mature technology, satellites can provide accurate location and timing signals at any time. In doors, however, the satellite signal becomes ineffective because of attenuation caused by buildings, interference during transmission, and multipath effect [3].

The indoor location technology can be divided into two types according to the signals used, i.e., wireless signal and visual information. The widely used wireless signals include WIFI [4], Bluetooth [5] and UWB (ultrawide band) [6], etc. Indoor location can be realized by measuring the distance, angle, or location fingerprint [3,4,5,6]. However, the cost to deploy base stations is comparatively high. Recently, with the development of Internet-of-Things technologies, some researchers devote themselves to indoor location using wireless local networks, which have already been deployed in many indoor environments [7,8]. Alhammadi et al. developed a three-dimensional indoor location system based on the Bayesian graphical model [9]. However, the average location error is more than 3 m and it can hardly be used in autonomous driving systems of intelligent mobile platforms. Moreover, there exists a multipath effect for the wireless communication signals. The visual location mainly includes SLAM (simultaneous location and mapping) [10] and location by visual tags [11]. SLAM can be applied to unknown environments by extracting and matching the texture feature automatically, but the algorithm has a higher computation complexity and its performance is also easily degraded by environmental disturbances. These pose great challenges on practical applications, especially for the scenario with a high demand on real-time and robust performance. Comparatively, the location algorithm using visual tags can achieve low computation complexity by prior storage of the feature and position information of tags. The selection of visual tags and pose calculation are important to location performance.

For the selection of visual tags, Ref. [12] uses special objects such as signs, desks, and doors, whose SURF feature is extracted to form the offline tag database. This method need not change the environment, but the information of texture features is still very large and it is also easily affected by environmental factors such as light. To reduce the storage requirement, only roof corners are considered in [13]. In [14], semantic segmentation technology is combined with the location algorithm for better performance, but much computing resource is required by the semantic segmentation process. No extra effort is needed to reconstruct the environment if using these already existing indoor objects. For the same type of object, however, their texture features are similar, which makes it difficult to discriminate different tags. To manage this problem, special tags are designed by coding the information of ID, coordinate, etc., in the tag to ensure its uniqueness [11]. On the contrary, the complexity of a QR code requires more time for detection, which even reaches hundreds of milliseconds [15]. Moreover, the texture information of complex QR codes is also easily degraded by environmental light and contaminants such as dust and dirt.

For the pose calculation, image matching and geometrical constraint are two widely used methods. In [11], the geometrical relationship of view between the tag image recorded in the database and that acquired in real time is used to calculate the position. Additionally, template matching is adopted in [16] to realize location by affine transformation. The fundamental of image matching is to find key texture features and establish their relationships between different images. This leads to a high degree of algorithm complexity. Comparatively, the computation of the algorithm using geometrical constraint is much less. It realizes location by geometrical relationship between the tags and sensors. The location precision reaches centimeter level by using a binocular camera for the detection of distance [13]. However, the computation resource required for the distance measurement by stereovision is very high [17]. In [18], the direct measured distance from multiple tags is used to construct overdetermined equations according to geometrical constraints, and the location is estimated by LSM (least square method). This approach has better comprehensive advantage on both location accuracy and computation complexity. When the redundancy of tags is not enough or the spatial distribution is special, however, the equations will become singular, which leads to a sharp increase in location error [19].

Considering the aforementioned problems, an indoor location system only using simple visual tags is designed for the autonomous driving system of intelligent mobile platforms. To manage the nonunique problem caused by the simple texture feature of tags, a fast query and matching algorithm is designed by using the FOV (field of view) of the camera and the tag azimuth. Furthermore, based on the relationship analysis between the spatial distribution of tags and location error, a pose and position estimation algorithm using WLSM (weighted least square method) is designed. Further, it works together with the interactive algorithm by the designed switching strategy to adapt the singular condition. A favorable balance has been achieved between the algorithm complexity and location accuracy by the presented techniques. The effectiveness of the location system designed herein is validated by both simulation and experiments, and compared with UWB, the location error is reduced greatly. The main contributions of this study are summarized as follows:

(1) A fast query and matching algorithm is designed to manage the nonunique problem of tags, which cannot be distinguished directly by their texture features.

(2) A cooperative strategy is proposed to combine WLSM and the interactive algorithm together to realize the estimation of pose and position with high accuracy under both general and singular conditions.

(3) An indoor location system only using simple visual tags is developed, which has the advantages of low consumption of computation resources, high accuracy, easy deployment, and robustness to environmental disturbances.

(4) The effectiveness of the proposed indoor location system is validated by both simulation and experiment tests.

The remainder of this paper is organized as follows: Section 2 describes the indoor location system design. The tag-matching strategy and the position calculation algorithm are introduced in Section 3 and Section 4, respectively; the effectiveness of the designed location system is validated by numerical simulations and comparative tests in Section 5; and Section 6 concludes the paper.

## 2. Indoor Location System Design

The visual tags and calculation algorithm of position are the two main parts in the designed system.

### 2.1. Simple Visual Tag Design

Based on the following reasons, the tags shown in Figure 1 are designed [20]:

(1) It is difficult to control the natural texture, which is also easily influenced by environmental factors;

(2) Tagging with a code is unique, but the detection algorithm is more complex and its sophisticated texture is also easily degraded;

(3) Cameras have strong capability to measure colors, and RGB are the three primary colors;

(4) It is easy to detect a circle, which also has the advantage of fine detection robustness and invariance from different views.

In this study, the tag is detected by the color and shape features [21], and the azimuth is measured by the tag’s center. In this way, better anti-interference capability can be achieved with less computation complexity. Only using these simple tag features, however, we cannot discriminate them directly. Accordingly, the pose and position cannot be calculated directly from the tag coordinates.

### 2.2. Location Algorithm with High Accuracy

To manage the nonunique problem of the simple visual tags, the location algorithm shown in Figure 2 is proposed after considering the advantages of the geometrical constraint method, i.e., high location accuracy and low algorithm complexity.

Considering that most intelligent platforms move indoors within the horizontal plane, the angle measurement method by vision is used to measure the tag position, as shown in Figure 3.

The optic center of camera (x,y), azimuth θ, and the tag position $\left(x_{i},y_{i}\right)$ satisfy [18,19]:(1)$$\tan\left(\theta +\alpha_{i}\right)=\frac{y_{i}-y}{x_{i}-x},\;\;\alpha_{i}=\arctan\frac{u_{i}-u_{0}}{f},\;\;i\in \Omega$$
where f is the focal distance and Ω is composed of the tag identifier, which can be observed by the camera. If the tag feature is unique and its coordinate is known, the pose and position can be solved from (1) directly, when the number of observed tags is no smaller than three. Information from more tags is beneficial to location accuracy and robustness [18]. Unfortunately, the designed tag only has color and shape features, which are nonunique. In the next section, a fast query and matching algorithm is designed by using the FOV of the camera and the tag azimuth. In this way, the matched tag can be found without searching the entire database. Then, with the coordinate of observed tags, a hybrid algorithm is presented in Section 3 to estimate the pose and position by combing WLSM and the interactive algorithm to create a favorable balance between algorithm complexity and location accuracy.

## 3. Matching of Tag

Considering the following reasons, the matching algorithm shown in Figure 4 is designed:

(1) The camera only can detect the tags in its FOV, which can be used to quickly search the database to determine the candidate tags for better efficiency;

(2) When the observed tag is consistent with a candidate tag, their azimuths should be exactly equal. This fact can be used to realize tag matching.

### 3.1. Determination of Candidate Tag

To determine the candidate tag quickly, the location result from the last cycle is used to estimate the current pose and position by DR (dead rocking) [22]. Then, according to the FOV model shown in Figure 5, the possible area where the candidate tag may exist can be determined. On the contrary, a tag locating outside this area cannot be the candidate tag.

In Figure 5, the coordinate of A is calculated by DR [22], and the coordinates of B and C, i.e., $\left(x_{B},y_{B}\right)$ and $\left(x_{C},y_{C}\right)$, are calculated by their geometrical relationship:(2)$$\left[\begin{array}{c} x_{B}\\ y_{B} \end{array}\right]=\left[\begin{array}{c} x_{A}\\ y_{A} \end{array}\right]+\frac{h}{\cos\gamma }\left[\begin{array}{c} \cos(\theta +\gamma )\\ \sin(\theta +\gamma ) \end{array}\right]\;\mathrm{and}\;\left[\begin{array}{c} x_{C}\\ y_{C} \end{array}\right]=\left[\begin{array}{c} x_{A}\\ y_{A} \end{array}\right]+\frac{h}{\cos\gamma }\left[\begin{array}{c} \cos\left(\theta -\gamma \right)\\ \sin\left(\theta -\gamma \right) \end{array}\right]$$
where h is the detection range and γ is the half horizontal view angle.

To improve the efficiency, the whole location area is rasterized to ensure that only one tag is allowed in one grid at most, and the stored tag information can be indexed by the grid number directly. According to the FOV model shown in Figure 5, the candidate tags are determined by the following steps:

Step 1: The grids covered by the camera FOV are determined by the coordinates of A, B, and C, which are calculated by (2).

Step 2: According to the grid number covered by the camera FOV, the possible tags are roughly selected from the database.

Step 3: For each tag $L_{i}$ selected in Step 2, whether it lies in FOV is judged accurately by the sign of pairwise dot products between the vectors, $\vec{L_{i}A}$, $\vec{L_{i}B}$, and $\vec{L_{i}C}$.

Step 4: The set of candidate tags $\Omega_{c}$ is composed of all tags whose pairwise dot products have the same sign. Additionally, the ideal azimuth of tags $\beta_{i},\;i\in \Omega_{c}$ is calculated by (1) according to the camera’s pose and position derived by DR.

### 3.2. Tag Matching by Azimuth

Considering the fact that when the candidate tag is consistent with the observed tag, their azimuths should be equal, we design the following strategy for tag matching:

Step 1: For any observed and unmatched tag, the error $e_{i}$ between its azimuth α and that of the candidate tag with the same color and shape feature is calculated by
(3)$$e_{i}=\left|\alpha -\beta_{i}\right|,\;i\in \Omega_{c}.$$

Step 2: If the minimum $e_{i}$ is smaller than the threshold, the candidate tag $L_{k},\;k=\arg\underset{i}{\min}e_{i}$ matches the observed tag. Otherwise, it is considered as a disturbance to be deleted.

Compared with the method using all feature information to perform the match by total optimization, this strategy can effectively avoid the matching error caused by the incomplete information of unobserved tags, which may be shaded by other objects, etc. With the aforementioned process, the coordinate of observed tags can be retrieved from the tag database quickly and accurately.

## 4. Pose and Position Calculation with High Accuracy

Theoretically, the pose and position can be calculated by (1) when the number of effective tags is not smaller than three [18]. In practical application, the following factors will degrade the location accuracy:

(1) The pixel error caused by factors such as environmental disturbances, camera pose, and tag position, is unfavorable for location accuracy.

(2) When solving the pose and position by (1), there exists matrix inversion. Under some special conditions, singularity causes the matrix to be invertible.

To manage these problems, a WLSM is designed to estimate the pose and position with high accuracy by analysis of the error transfer characteristics using (1), and furthermore, a cooperation strategy is presented to schedule WLSM and the interactive algorithm to adapt to the singular condition.

### 4.1. Analysis of Error Transfer Characteristic

Considering the fact that the pose and position can be calculated with three effective tags, the following error transfer characteristic is obtained by the sensitive analysis of (1):(4)$$\left[\begin{array}{c} d\alpha_{i}\\ d\alpha_{j}\\ d\alpha_{k} \end{array}\right]=G\left[\begin{array}{c} dx\\ dy\\ d\theta \end{array}\right],\;\;G=\left[\begin{array}{ccc} \frac{\sin\left(\alpha_{i}+\theta \right)}{r_{i}} & \frac{-\cos\left(\alpha_{i}+\theta \right)}{r_{i}} & -1\\ \frac{\sin\left(\alpha_{j}+\theta \right)}{r_{j}} & \frac{-\cos\left(\alpha_{j}+\theta \right)}{r_{j}} & -1\\ \frac{\sin\left(\alpha_{k}+\theta \right)}{r_{k}} & \frac{-\cos\left(\alpha_{k}+\theta \right)}{r_{k}} & -1 \end{array}\right].$$
where $\alpha_{i}$ and $r_{i}$ are the measured azimuth of the tag $L_{i}$ and its distance to the camera. When the camera and the tags are on the same circle, as shown in Figure 6, we have the following equations according to Ptolemy theorem and law of sines:(5)$$r_{i}d_{jk}+r_{k}d_{ij}=r_{j}d_{ik},\;\;\frac{d_{ij}}{\sin\left(\alpha_{i}-\alpha_{j}\right)}=\frac{d_{jk}}{\sin\left(\alpha_{j}-\alpha_{k}\right)}=\frac{d_{ik}}{\sin\left(\alpha_{i}-\alpha_{k}\right)}.$$

Under this condition, it is known from (5) that |G|=0 and the position equations become ill-conditioned. This implies that a small error in the measured azimuth will result in significant deviation in the solved pose and position. Accordingly, the following cooperation strategy is designed. When the condition number of G is smaller than the threshold, the pose and position is calculated by WLSM as it is given in Section 4.2. Otherwise, the following interactive searching algorithm is used:

Step 1: The possible azimuth range is discretized according to the requirement of accuracy.

Step 2: For each discretized azimuth, the camera coordinate can be calculated by the following equation according to the information of any two tags:(6)$$\begin{array}{c} x=\frac{\left(t_{i}t_{\theta }-1\right)\left(t_{j}t_{\theta }-1\right)\left(y_{i}-y_{j}\right)+\left(t_{j}t_{\theta }-1\right)\left(t_{\theta }+t_{i}\right)x_{i}-\left(t_{i}t_{\theta }-1\right)\left(t_{\theta }+t_{j}\right)x_{j}}{\left(t_{\theta }^{2}+1\right)\left(t_{j}-t_{i}\right)}\\ y=\frac{\left(t_{\theta }+t_{i}\right)\left(t_{\theta }+t_{j}\right)\left(x_{i}-x_{j}\right)+\left(t_{i}t_{\theta }-1\right)\left(t_{\theta }+t_{j}\right)y_{i}-\left(t_{j}t_{\theta }-1\right)\left(t_{\theta }+t_{i}\right)y_{j}}{\left(t_{\theta }^{2}+1\right)\left(t_{i}-t_{j}\right)}\\ t_{i}=\tan\alpha_{i},\;\;t_{j}=\tan\alpha_{j},\;\;t_{\theta }=\tan\theta \end{array}$$
where $\left(x_{i},y_{i}\right)$ and $\alpha_{i}$ are the stored position of tag $L_{i}$ and its measured azimuth, respectively, and θ is the discretized azimuth.

Step 3: For each discretized azimuth, the variance of camera’s coordinate is calculated. The coordinate with the smallest scattered degree and the corresponding azimuth are then selected as the effective pose and position value.

### 4.2. Position Estimation by Weighted Least Square Method

Under the noncircular conditions, the pose and position can be solved by every three tags. It has the advantage of fine real-time performance and good robustness to noise to solve the aforementioned overdetermined equations by LSM [23]. However, the location error from different tag groups is influenced by many factors, such as environments, image resolution, etc. To further improve the location accuracy, a weighted least square estimation algorithm is presented by designing the evaluation index of solution quality.

Taking the estimation of θ as an example, the following equation is derived for any group of tags, $L_{i},L_{j}$, and $L_{k}$, from the geometrical constraint described by (1) [18]:(7)$$\begin{array}{c} \sigma_{t_{\theta }}^{2}=\frac{\left[\begin{array}{ccc} \frac{\partial t_{\theta }}{\partial t_{i}} & \frac{\partial t_{\theta }}{\partial t_{j}} & \frac{\partial t_{\theta }}{\partial t_{k}} \end{array}\right]{\left[\begin{array}{ccc} \frac{\partial t_{\theta }}{\partial t_{i}} & \frac{\partial t_{\theta }}{\partial t_{j}} & \frac{\partial t_{\theta }}{\partial t_{k}} \end{array}\right]}^{T}}{f^{2}}\sigma_{u}^{2},\\ t_{\theta }=\frac{\left(t_{k}-t_{i}\right)\left(-y_{j}+t_{j}x_{k}+y_{i}-t_{i}x_{i}\right)-\left(t_{j}-t_{i}\right)\left(-y_{k}+t_{k}x_{k}+y_{i}-t_{i}x_{i}\right)}{\left(t_{k}-t_{i}\right)\left(-x_{j}-t_{j}y_{j}+x_{i}+t_{i}y_{i}\right)-\left(t_{j}-t_{i}\right)\left(-x_{k}+t_{k}y_{k}+x_{i}-t_{i}y_{i}\right)} \end{array}$$
where $\sigma_{t_{\theta }}^{2}$ and $\sigma_{u}^{2}$ are the variances of the tangent value of θ and the pixel coordinate u. Equation (7) establishes the relationship between the pixel error of the tag center and the azimuth error.

When using WLSM, the maximum likelihood solution can be obtained by setting the weight as the reciprocal of error variance [23]. From the error relationship described by (7), the WLSM used to optimally estimate the azimuth is designed as:(8)$$\begin{array}{c} \theta =\mathrm{atan}t_{\theta },\;\;t_{\theta }={\left(A^{T}WA\right)}^{-1}A^{T}WB,\\ A={\left[\begin{array}{ccc} a_{1} & a_{2} & \cdots \end{array}\right]}^{T},B={\left[\begin{array}{ccc} b_{1} & b_{2} & \cdots \end{array}\right]}^{T},W=\mathrm{diag}\left(1/\omega_{1},1/\omega_{2},\cdots \right),\\ a_{n}=\left(t_{k}-t_{i}\right)\left(x_{i}-x_{j}+\left[\begin{array}{cc} t_{i} & -t_{j} \end{array}\right]\left[\begin{array}{c} y_{i}\\ y_{j} \end{array}\right]\right)-\left(t_{j}-t_{i}\right)\left(x_{i}-x_{k}+\left[\begin{array}{cc} t_{i} & -t_{k} \end{array}\right]\left[\begin{array}{c} y_{i}\\ y_{k} \end{array}\right]\right),\\ b_{n}=\left(t_{k}-t_{i}\right)\left(y_{i}-y_{j}+\left[\begin{array}{cc} -t_{i} & t_{j} \end{array}\right]\left[\begin{array}{c} x_{i}\\ x_{j} \end{array}\right]\right)-\left(t_{j}-t_{i}\right)(y_{i}-y_{k}+\left[\begin{array}{cc} -t_{i} & t_{k} \end{array}\right]\left[\begin{array}{c} x_{i}\\ x_{k} \end{array}\right],\\ \omega_{n}=\left[\begin{array}{ccc} \frac{\partial t_{\theta }}{\partial t_{i}} & \frac{\partial t_{\theta }}{\partial t_{j}} & \frac{\partial t_{\theta }}{\partial t_{k}} \end{array}\right]{\left[\begin{array}{ccc} \frac{\partial t_{\theta }}{\partial t_{i}} & \frac{\partial t_{\theta }}{\partial t_{j}} & \frac{\partial t_{\theta }}{\partial t_{k}} \end{array}\right]}^{T}. \end{array}$$

## 5. Validation and Analysis

In this section, following validation, the effectiveness of the proposed method by numerical simulations is further compared with UWB by experiments.

### 5.1. Simulation Validation and Analysis

The location scenarios shown in Figure 7 are designed to indicate the effectiveness of the designed cooperation strategy and WLSM. Under condition (a), all tags locate on the same circle, which is used to validate the cooperative strategy. Under condition (b), tags 1, 2, and 3 are on the same circle, but tag 4 is not, and the camera moves along the circle. This condition is used to validate that the designed WLSM can attenuate the negative influence from the unfavorable tag group. The scenario is simulated in Prescan and the designed location algorithm runs in Matlab. Prescan provides the integrated interfaces for cosimulation with Matlab [24]. During the simulation, the time step is set to be 0.03 s, the radius of the circular trajectory is 5 m, and the moving speed of the camera is about 0.5 m/s.

Shown in Figure 8 are the location results when the camera moves from the outside of the circle along the arrow direction shown in Figure 7a. At about 1.3~1.9 s, the camera and tags are on the same circle. The overdetermined Equation (1) becomes singular and the location error of LSM increases quickly to a large value. On the contrary, the proposed cooperative algorithm can detect the singularity and switch to the interactive algorithm as soon as possible. Accordingly, the location accuracy can be ensured during the whole simulation process.

Since pose and position can be solved with any 3 tags, under condition (b), 4 tags can generate 4 combinations with every 3 tags. The combination of tags 2, 3, and 4 is denoted by group 1, the combination of tags 1, 2, and 3 is denoted by group 2, the combination of tags 1, 3, and 4 is denoted by group 3, and the combination of tags 1, 2, and 4 is denoted by group 4. Among them, the tags in group 2 and the camera are always on the same circle. The solution quality for group 2 is worst. It is found from Figure 9a that the contribution from group 2 on the location results is almost zero. This shows that the designed index to evaluate the solution quality given in Section 4.2 can measure the confidence level of results from different tag groups. Accordingly, the presented WLSM using this index as the weight can successfully attenuate the negative influence from unfavorable tag groups. Compared with LSM, it can be found from Figure 9c,d that the maximum errors of azimuth, longitudinal, and lateral location results are reduced by 89.47%, 51.61%, and 88.78%, respectively.

### 5.2. Comparative Test of Location Accuracy

To further test the location accuracy, the designed location system is compared with UWB under the indoor condition shown in Figure 10 considering the following reasons: (1) Methods such as ORB-SLAM have a high requirement for computation resources. (2) The visual methods using complex texture features can hardly work under such disturbed conditions because some main features are covered or light obviously changes. (3) The average error of the methods using location fingerprint of wireless signals is up to several meters. (4) Comparatively, UWB has the advantages of low computation complexity, high accuracy, and robustness to environmental disturbances such as light and dynamic covering of objects. The room size is about 4.1 m × 5.2 m, where there are several desks, which may degrade the location performance of UWB. About 20 tags are stuck on the walls around the room, and 4 UWB stations are used. To simulate the environmental disturbances, the light is turned on/off and the tags are covered randomly by some objects during the test.

During the test, the location algorithm runs in a Raspberry 4b board with the rate 30 fps. The board with the camera is held by hand and moves along the defined trajectory as best we can with the speed of about 0.5 m/s. The main test equipment with their technical parameters is listed in Table 1. The camera is used to detect the tag, and in addition to the algorithm, the location results from both the proposed method and UWB are recorded by the Raspberry 4b.

The location results are shown in Figure 11 when the camera moves along the defined trajectory.

The average location error of the results is shown in Table 2. Compared with UWB, the longitudinal and lateral location error of the designed system is reduced by 65.93% and 69.75%, respectively.

Since it is difficult to control the node to move along the trajectory accurately and strictly, the statistical error in Table 2 is larger than their actual values. Ten test points are selected where the location results are measured statically for accuracy, and the results are shown in Figure 12. On the whole, the average location error of UWB is more than 13.5 cm, reaching a maximum of 49 cm at position (1.61 cm, −0.50 cm). The wireless signal of UWB is reflected by the objects in the room, which degrades the location performance of UWB by multipath effect. Even though with the simulated environmental disturbances of light and covering, the average error of the proposed location system is about 5.5 cm and the maximum error is only 9 cm. The location accuracy is obviously improved by the proposed techniques in this study.

## 6. Conclusions

To realize indoor locations with high accuracy, low cost, and robustness, an indoor location system is designed in this study by presenting a fast tag-matching strategy and cooperative-solving algorithm. The system locates according to simple visual tags, which are robust to environmental disturbances and easily deployed. The cooperative-solving algorithm for estimation of pose and position can work under cocircular conditions, and compared with LSM, the maximum errors of azimuth, longitudinal, and lateral location results are reduced by 89.47%, 51.61%, and 88.78%, respectively. The average location error for the designed location system under disturbed environments is about 5.5 cm, and the location accuracy is higher than UWB by 62%.

## Figures and Tables

**Figure 1 sensors-23-01597-f001:**
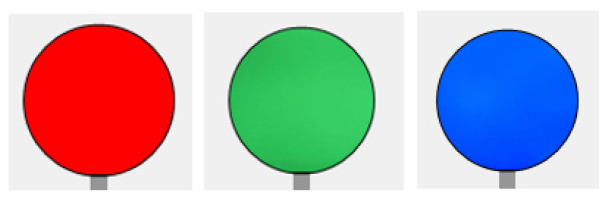
Simple visual tags.

**Figure 2 sensors-23-01597-f002:**
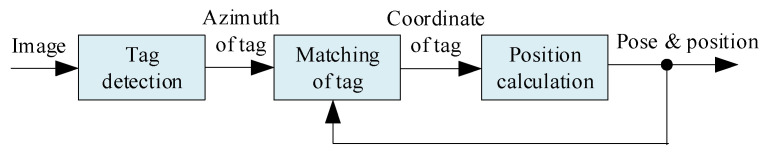
Location algorithm with high accuracy.

**Figure 3 sensors-23-01597-f003:**
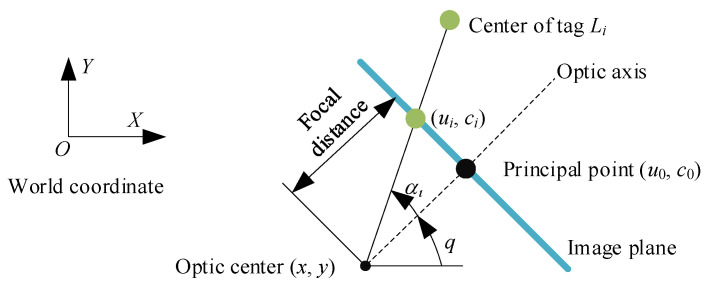
Angle measurement by vision.

**Figure 4 sensors-23-01597-f004:**
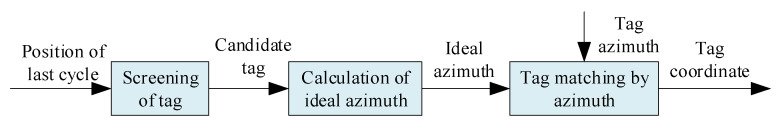
Matching process for observed and stored tags.

**Figure 5 sensors-23-01597-f005:**
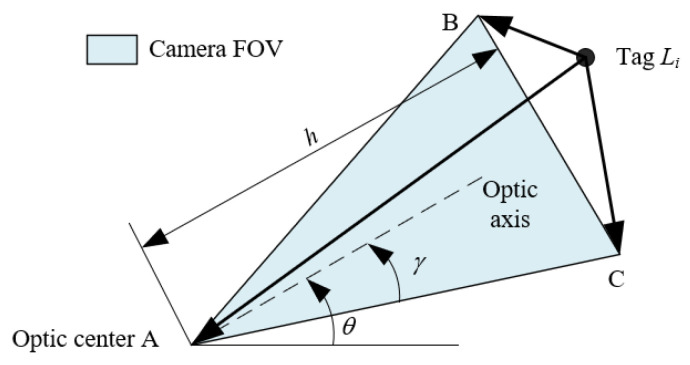
FOV model of camera.

**Figure 6 sensors-23-01597-f006:**
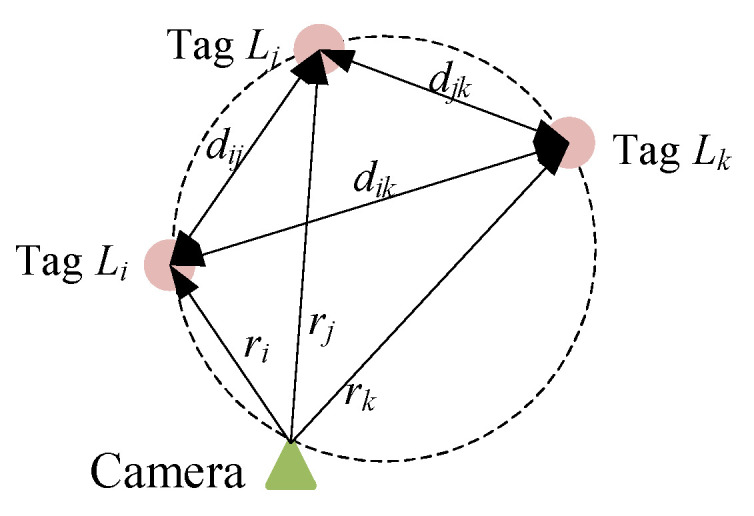
Schematic for cocircular condition.

**Figure 7 sensors-23-01597-f007:**
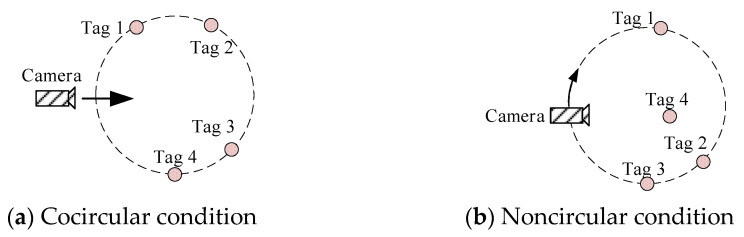
Simulation conditions.

**Figure 8 sensors-23-01597-f008:**
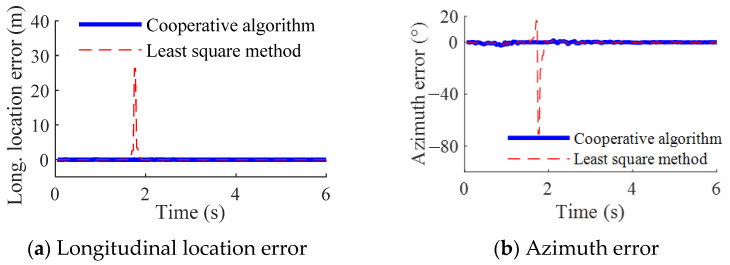
Location results under condition (a).

**Figure 9 sensors-23-01597-f009:**
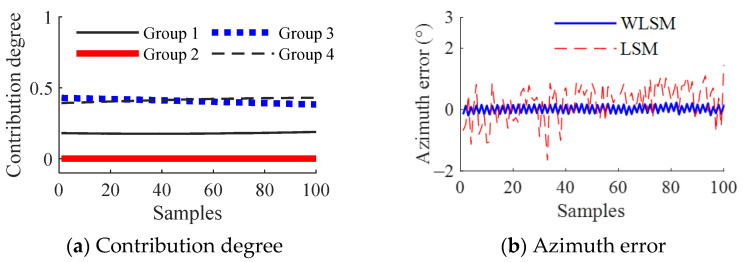
Location results under condition (b).

**Figure 10 sensors-23-01597-f010:**
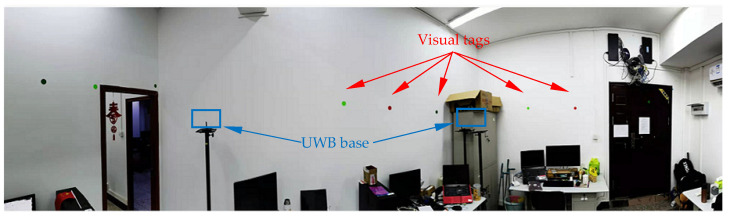
Indoor location scenario.

**Figure 11 sensors-23-01597-f011:**
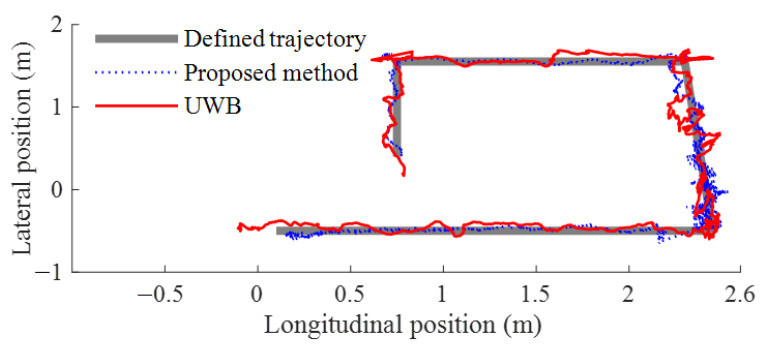
Comparative test results.

**Figure 12 sensors-23-01597-f012:**
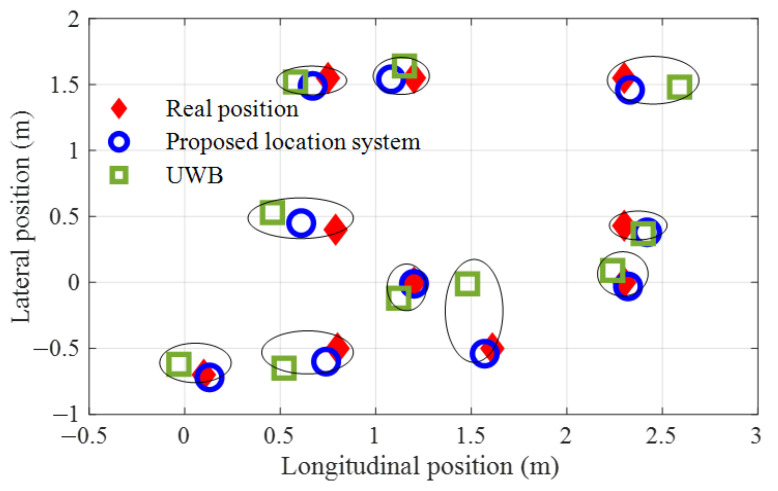
Comparative test results under statistical conditions.

**Table 1 sensors-23-01597-t001:** Test equipment and technical parameters.

Equipment	Parameter	Value
Camera	Resolution	1920 × 1080
	Frame rate	30 fps
	Field of view	78 degree
	Focal length	747.1 pix
Raspberry 4b	CPU	BCM2837B0SOC
	Number of cores	4
	Basic frequency	1.4 GHz
	RAM	1 G
UWB	Location precision	10 cm
	Communication range	≤130 m
	Update frequency	≤50 Hz

**Table 2 sensors-23-01597-t002:** Statistical location errors.

Average Error	Long. Location (cm)	Lat. Location (cm)	Azimuth (°)
UWB	36.1	32.4	/
Proposed location system	12.3	9.8	0.88

## Data Availability

The data are available upon request to the corresponding author.

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
