# Peer review of "Indoor Location Technology with High Accuracy Using Simple Visual Tags"

_sensors, 2023, doi:10.3390/s23031597_

Round 1

Reviewer 1 Report

It is an interesting study, and the paper is well organized, but below comments need to be addressed.

1. The figure number is missing in line 272.

2. UWB location results are not satisfactory, and using the UWB results is not necessary, and I would suggest the authors using other algorithm for comparison.

3. In the test, defined trajectory is used as the groundtruth, but it is better to use true trajectory as the groundtruth.

4. In the tag design, it is claimed that rgb color is needed, but in Fig. 11, only green tag is used. An explaination is needed. 

Reviewer 2 Report

 This paper designed an indoor location system using simple visual tags considering the accuracy and computation complexity. The paper has limited contributions and is well-organized. I have several comments and suggestions as follows:

- In the introduction, the papers' contributions are not well discussed, I suggest listing down the contribution of the paper into several points for more clarity. 

- I suggest adding a new section for related work and including the following works: An overview of indoor localization technologies: Toward IoT navigation services", " A review on wireless emerging IoT indoor localization" and "A three-dimensional pattern recognition localization system based on a Bayesian graphical model".

- In section 5, more details are missing such as simulation settings and parameters have not been discussed. In addition, the strategy of grouping tags is not mentioned. The authors simply say that groups 1, 2, 3, and 4 without mentioning how they did grouping these tags.

        -The experimental part also needs more discussion such as experimental setup and settings, testbed dimension, data collection, number of tags, and data analysis. Besides, the results need more discussion and further analysis. 

       - Incomplete sentence on page 8, lines 271 and 272. "To further test the location accuracy, the designed location system is compared with 271 UWB under the indoor condition shown in."

- The conclusion should be rewritten in a proper way. 

Round 2

Reviewer 1 Report

The authors have addressed my concerns and I am satisfied with the revised paper.

Reviewer 2 Report

Thanks for addressing my concerns, I do not any have any further comments.